# Nanoscale reshaping of resonant dielectric microstructures by light-driven explosions

Maxim R. Shcherbakov [1,2] ✉, Giovanni Sartorello [3,4], Simin Zhang [5], Joshua Bocanegra [1,6], Melissa Bosch[3,7], Michael Tripepi [8,9], Noah Talisa[9,10], Abdallah AlShafey[9], Joseph Smith [11], Stephen Londo[12], François Légaré[12], Enam Chowdhury[5,9,13] & Gennady Shvets[3]

Femtosecond-laser-assisted material restructuring employs extreme optical intensities to localize the ablation regions. To overcome the minimum feature size limit set by the wave nature of photons, there is a need for new approaches to tailored material processing at the nanoscale. Here, we report the formation of deeply-subwavelength features in silicon, enabled by localized laser-induced phase explosions in prefabricated silicon resonators. Using short trains of mid-infrared laser pulses, we demonstrate the controllable formation of high aspect ratio (>10:1) nanotrenches as narrow as $\sim\lambda/\mathbf{80}$. The trench geometry is shown to be scalable with wavelength, and controlled by multiple parameters of the laser pulse train, such as the intensity and polarization of each laser pulse and their total number. Particle-in-cell simulations reveal localized heating of silicon beyond its boiling point and suggest its subsequent phase explosion on the nanoscale commensurate with the experimental data. The observed femtosecond-laser assisted nanostructuring of engineered microstructures (FLANEM) expands the nanofabrication toolbox and opens exciting opportunities for high-throughput optical methods of nanoscale structuring of solid materials.

The progress in light-based material processing[1–4], fabrication[5], surgery[6], and lithography[7] is inherently limited by the spatial scale of laser-matter interactions. The diffraction limit dictates that a conventional lens cannot localize light to a spatial scale finer than $\lambda_L/2n$, where $\lambda_L$ is the wavelength of light in a vacuum, and $n$ is the refractive index of the surrounding material[8], thus preventing deeply-subwavelength material patterning using free-space optics. Employing nonlinear optical processes, such as multiphoton absorption[9,10],

can potentially shrink the modified region by a factor of $\sim\sqrt{p}$, where $p \le 5$ is the order of the nonlinear process[2], although the accuracy of this estimate has been questioned[11]. Remarkable progress in highly-controllable sub-micron precision laser ablation and nanoparticles formation has been made since the advent of femtosecond-laser systems, primarily in the context of planar dielectric and semiconductor materials[12–17]. Several non-thermal mechanisms of surface modification have been identified[13–18]; all the mechanisms rely on rapid surface-

[1]Department of Electrical Engineering and Computer Science, University of California, Irvine, CA 92697, USA. [2]Beckman Laser Institute and Medical Clinic, University of California, Irvine, CA 92612, USA. [3]School of Applied and Engineering Physics, Cornell University, Ithaca, NY 14850, USA. [4]Cornell NanoScale Science and Technology Facility, Cornell University, Ithaca, NY 14853, USA. [5]Department of Material Science and Engineering, The Ohio State University, Columbus, OH 43210, USA. [6]Department of Physics, University of California, Irvine, CA 92697, USA. [7]Department of Physics, Cornell University, Ithaca, NY 14850, USA. [8]Physics Department, Hillsdale College, Hillsdale, MI 49242, USA. [9]Department of Physics, The Ohio State University, Columbus, OH 43210, USA. [10]Johns Hopkins University Applied Physics Laboratory, Laurel, MD 20723, USA. [11]Physics Department, Marietta College, Marietta, OH 45750, USA. [12]Advanced Laser Light Source (ALLS) at Centre Énergie Matériaux Télécommunications, Institut national de la recherche scientifique, Varennes, Québec J3X 1P7, Canada. [13]Department of Electrical and Computer Engineering, The Ohio State University, Columbus, OH 43210, USA. ✉e-mail: maxim.shcherbakov@uci.edu

localized generation of high-density hot electron-hole (e-h) plasma via highly nonlinear optical processes; those are frequently followed by impact ionization. The key practical limitation of laser-based modification of planar surfaces is a fairly limited repertoire of possible nano-patterns, such as single and arrayed craters[14], as well as submicron particles[15]. Deeply-subwavelength nanoholes have been machined into dielectric substrates by operating at the laser fluences just above the ablation threshold[1], albeit under exceedingly strict laser stability requirements[11].

An alternative approach to subwavelength nanostructuring employs resonant plasmonic nanostructures capable of overcoming the diffraction limit by funneling laser fields into subwavelength hot spots[19,20], thereby enabling super-resolution[7] or laser-induced nanoscale particle reshaping[21]. More recently, all-dielectric nanoresonators[22–24] supporting subwavelength hot spots emerged as a powerful platform for boosting nonlinear light-matter interactions[25–27]. Even more remarkably, plasma generation between two halves of a grape in a microwave oven – a mainstay of science-fair projects – has been recently shown to be related to Mie resonances of high-index hemispheres[28]. Translating this concept of hot-spot-assisted subwavelength material modification by a resonant all-dielectric structure into the optical domain would enable high-throughput nanoscale "machining" of solid materials that could significantly enhance current nanofabrication approaches.

Here, we achieve deeply-subwavelength laser-induced modification of all-dielectric microstructures by tailored explosions of the constituent material – silicon, in our case – induced by resonantly-tuned mid-infrared laser pulses. By designing an array of M-shaped microparticles to resonate approximately at the laser pulse frequency, we achieve controlled modifications of the micron-scale prefabricated resonators by femtosecond-laser pulses. The emerging nanoscale features include narrow (down to nearly $\lambda_L/80$), high aspect ratio (over 10:1) trenches. We study the trench formation as a function of the number, intensity, and polarization of the laser pulses interacting with Si resonators. For optimal pulse intensity and polarization, we find that the deep trench – spanning the entire micron-scale height of the

resonator – propagates laterally through the structure at a rate of approximately $v_{tr} \sim 30$ nm per laser pulse.

The volumetric nature of the discovered laser-driven explosion of nanostructures makes it conceptually distinct from the commonly studied surface ablation of planar dielectric and semiconductor surfaces[1,12–17,29] while maintaining some similarities. Just like all ultrafast laser ablation techniques, FLANEM relies on the rapid generation of electron-hole plasma via multiphoton or tunneling ionization. The plasma is then heated by the laser field through a variety of mechanisms (e.g., electron-lattice and electron-hole scattering[30,31], non-resonant Brunel[32], and resonant[33] absorption mechanisms), and finally transfers its energy to the ions via either Coulomb or phase explosions[14]. There are, however, several critical distinctions between the FLANEM mechanism described below and its flat-surface counterparts. First, the efficiency of light coupling into the photo-generated e-h plasma is limited by its density $n_p$ as soon as the plasma frequency $\omega_p = \sqrt{n_p q_e^2 \varepsilon_0 / m_{eff}}$ satisfies $\omega_p \gg \omega_L$, where $m_{eff} \approx 0.3 m_e$ is the effective charge carrier mass in Si[31], $q_e$ is the electron charge, $\varepsilon_0$ is the vacuum permittivity, and $\omega_L = 2\pi c / \lambda_L$. This coupling reduction does not occur in a three-dimensional (3D) dielectric microstructure exemplified by Fig. 1a because the optical field does not need to pass through the region of the high-density e-h plasma – the microstructure is electrically polarized from its side by the incident laser field. Similarly, FLANEM is distinct from near-field ablation enabled by subwavelength plasmonic nanoparticles[34,35]. The linear nature of hot-spot formation in metallic particles reduces the expected localization of ablated material, reaching inscribed features to $\sim \lambda_L/10$ in lateral dimensions, whereas the highly nonlinear nature of FLANEM enables almost an order of magnitude improvement in this figure.

Second, the high-Q nature of the dielectric microresonator enables the electric field to build up over time. As a result, the metallization of Si inside the small portion of a microresonator takes place when the optical energy of the laser pulse is already coupled into it. Finally, the resonant enhancement of the optical field inside the microstructure enables the production of dense (on the order of $n_p \sim 10^{22} cm^{-3}$) e-h plasma at a relatively small incident threshold

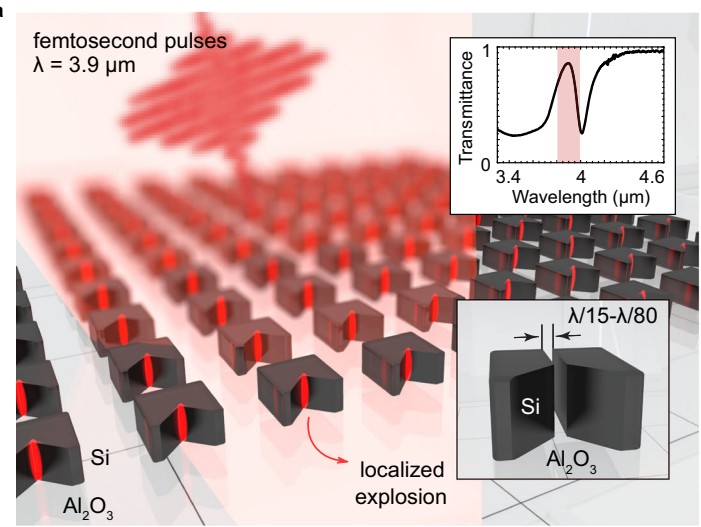

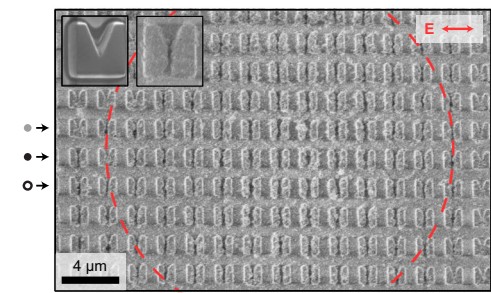

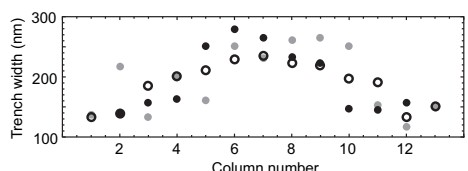

**Fig. 1 | FLANEM in resonant M-shaped microresonators: deep-subwavelength nanotrench formation.** a Schematic of the nanotrench formation by localized explosions: (i) the resonator is irradiated by a powerful mid-infrared femtosecond-laser pulse; (ii) a local-field hot spot is formed, inducing high-density high-energy electron-hole plasma generation; (iii) at a critical plasma density, breakdown occurs, forming a subwavelength trench. Upper inset: experimental transmittance spectrum exhibiting a resonant dip; red rectangle: fwhm of the pulse's intensity spectrum. b An array of M-shaped resonators after irradiation by 100 pulses at a peak fluence of $J_1 \sim 0.19 J cm^{-2}$, showing trench formation. Dashed red circle: fwhm of the laser beam spot. The insets: resonators before (left) and after (right) irradiation. c Spatial distribution of the trench widths across SEM image in (b) showing a correlation between the trench width and pulse intensity. Three rows are shown, marked with grey dots, black dots, and open circles in (b).

fluence of $J_{thr} \sim 0.1$ J cm$^{-2}$ : almost an order of magnitude lower than typically used for multiphoton laser patterning of flat Si surfaces[36]. Supported by particle-in-cell (PIC) simulations of local electronic plasma density and lattice ion temperature evolution, we conjecture that the resulting nanoscale ablation of Si is attributed to highly localized Coulomb or phase explosions, followed by rapid ejection of superheated material. The FLANEM process can be extended to a wide variety of wavelengths and spatial scales, and tailored local fields will enable new modalities of light-based material nanostructuring.

## Results

### Subwavelength trench formation observation

The formation of a nanotrench in an M-shaped dielectric resonator is conceptualized by Fig. 1a. Localized high-density e-h plasma is generated at the "hot spots" of prefabricated microresonators illuminated by an intense mid-infrared (MIR) femtosecond-laser pulse, resulting in rapid heating and ablation of silicon in the areas of the highest local-field intensities. The microresonator arrays are fabricated on a silicon-on-sapphire wafer (silicon layer thickness: $h = 600$ nm); see "Methods" for the fabrication procedure. The resonator arrays exhibit an optical resonance at a wavelength of around $\lambda_{res} = 4$ μm as evidenced by Fourier transform infrared (FTIR) spectroscopy of the sample probed with a beam polarized along the bottom side of the M-shaped resonators. The resonators have a fundamental resonant magnetic-dipole mode at the wavelength of the laser source, as evidenced from the local-field map calculated in the mid-plane of the resonator using COMSOL software package. The resonance is experimentally manifested as a transmission dip in the FTIR spectrum shown in the upper inset of Fig. 1a. A scanning electron microscope (SEM) image of a typical resonator before irradiation is shown in the inset of Fig. 1b.

Variable-count ($N \le 100$) trains of $\tau_L \approx 200$fs-long linearly-polarized MIR pulses with fixed central wavelength $\lambda_L = 3.9$ μm and per-pulse pulse energy of up to $P_L = 6$ μJ from an optical parametric amplifier (OPA) were focused down a spot with a diameter of $D_L = 36$ μm: see the schematic of the experimental setup in Fig. 2b showing how $N$ is controlled by a mechanical shutter, and their polarization direction $\theta$ is set with respect to the horizontal base of the microresonators. To localize the material removal to deeply-subwavelength hot spot regions, the resulting peak fluence of up to $J_{max} \approx 0.28$ J cm$^{-2}$ and an average intensity of up to $I_{max} \approx 1.4$ TW cm$^{-2}$ were intentionally chosen below the bulk damage threshold $J_{bulk} \approx 0.5$ J cm$^{-2}$ of pristine silicon of in this spectral range[36] (see "Methods" for the details of the experimental setup). Since the energy of the photon $\hbar\omega_L = 0.32$ eV is much smaller than the bandgap of silicon $E_g = 1.14$ eV, the process of e-h plasma generation – a combination of multiphoton ionization and tunneling – is highly nonlinear and, therefore, spatially localized[1,11]. Such localization can be viewed as a cascade of the relevant spatial scales: from the laser spot size (multiple microns) to the hot spot of the microresonator (hundreds of nanometers), to the region of nonlinear e-h plasma generation (tens of nanometers).

A typical *post mortem* SEM image of a sample irradiated with $N = 100$ pulses is shown in Fig. 1b, where the dashed circle denotes the intensity full width at half maximum (fwhm) of the MIR beam with the peak on-axis fluence of $J_1 \sim 0.19$ J cm$^{-2}$. As a result of laser irradiation, nanoscale trenches are formed in the microresonators that, roughly, fall within the dashed red circle in Fig. 1b. The average width $w$ of the resulting nanotrenches were recovered from the SEM images and plotted as a function of the horizontal coordinate (labeled in Fig. 1c as the "resonator number") running across the irradiating beam spot; this

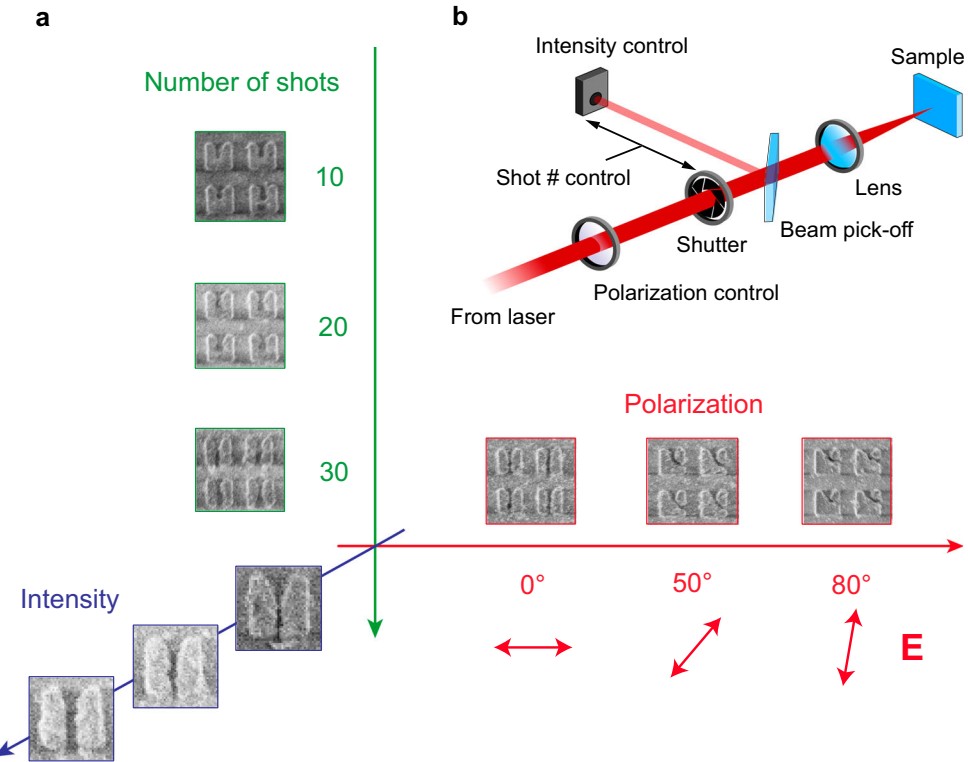

**Fig. 2 | Control of the nanotrench formation using polarization, peak intensity, and the length of the train of MIR pulses. a** Examples of trench control through polarization, peak intensity, and the length of the train of MIR pulses. The trench is seeded at the apex of the M-shaped resonator, increases in length with the number of pulses in the train (vertical axis). The direction of the trench formation is controlled by the polarization angle $\theta$ of the MIR pulse, showcased by three examples at $\theta = 0°, 50°$ and $80°$ (horizontal axis). The trench widens with peak intensity (out-of-plane axis). Each resonator is approximately $1.2 \mu$m in width. **b** Setup schematic. Polarization of 200-fs pulses from the OPA is controlled by rotating a half-waveplate. The length of the pulse train is controlled by a mechanical shutter synchronized with the laser source.

coordinate acts as a proxy for the spatially-varying local fluence[37]. We find that $115\,nm < w < 280\,nm$ on a given sample (or $\lambda_L/34 < w < \lambda_L/14$, respectively), with the largest nanotrenches observed near the laser beam center.

### Control of nanoscale reshaping

The results of our measurements compiled in Fig. 2a indicate that the process of nanotrench formation can be controlled by the parameters of the irradiating beam, such as the length $N$ of the pulse train delivered to the sample (vertical axis), as well as the peak intensity (out-of-page axis) and the polarization direction $\theta$ (horizontal axis) of the individual MIR pulses. The SEM images of the samples irradiated with $N = 10, 20, 30$ pulses shown in Fig. 2a indicate that, while $N = 10$ or $N = 20$ is a sufficient number of pulses for initiating the nanotrench formation at the apex of the M-shaped microresonator, the nanotrench is fully formed only starting with $N = 30$ pulses. The average lateral trench propagation speed is estimated at rate of $v_{abl} \approx 30\,nm$ per pulse, consistent with past observations in $SiO_2$[38]. Notably, the ablation process is fully volumetric, i.e., the trench spans the entire $h = 600\,nm$ thickness of the microresonator.

Remarkably, the orientation of the nanotrench is controlled by the polarization of the MIR beam. Figure 2a shows examples of $N = 100$-pulse trench formation for three polarization angles, $\theta = 0°$, $50°$, $80°$, where $\theta = 0°$ corresponds to the horizontal polarization direction parallel to the bottom side of the M-shaped resonator. Our PIC simulations (see below) show that changing the laser polarization direction from horizontal ($\theta = 0°$) to tilted ($\theta \neq 0°$) results in an asymmetric distribution of the free carriers inside the microstructure and, therefore, lays the ground for asymmetric ablation of the target material. The precise reasons for this asymmetry as still under investigation, but we speculate that they could be related to asymmetric launching of the escaped electrons back into the target via Brunel mechanism[32]. Alternatively, the asymmetry may be assisted by stress-induced breaking[39] along one of the crystallographic axes of Si driven by Griffith's criterion of crack propagation[40]. We envision that complex polarization-driven fields and pulse count control can enrich the palette of possible reshaping outcomes.

We observe the scalable nature of nanotrench formation. Figure 3 shows the results of the same experiment repeated at upscaled conditions, where the pump wavelength ($\lambda_L \approx 6.9\,\mu m$) and the dimensions are proportionally increased to maintain the resonance around the same wavelength. In Fig. 3a, the transmittance shows an out-of-plane magnetic-dipole mode at $\lambda_{res} \approx 7\,\mu m$, with the absolute field strength map shown in the inset. In Fig. 3b, a shot-controlled behavior is seen,

with the trench being seeded at $N = 10$ shots, halfway through the resonator at $N = 30$ shots, and fully trenched at $N = 50$ shots. Expectedly, pulse fluence can control the trench width as well, going from 150 nm at $0.15\,J\,cm^{-2}$ ($\approx \lambda_L/50$) to 550 nm at $1.2\,J\,cm^{-2}$.

## Discussion

The reshaping mechanism in designer microresonators can be explained through a simple three-step model. In Fig. 4a, the resonant absorption of the MIR light through multiphoton and tunnel ionization (I) leads to local-field redistribution predominantly toward the apex of the resonator (II), which leads to localized material ablation (III). Figure 4b shows the local-field distribution in the midplane ($z = d/2$) of the resonator calculated using a COMSOL solver before the pulse arrival (left: $\omega_p = 0$) and during the pulse (right: $\omega_p \neq 0$). For simplicity, we assumed that $\omega_p = 10^{15}\,rad\,s^{-1}$ (corresponding to $n_p \sim 10^{20}\,cm^{-3}$) and spatially uniform. While the unperturbed ("cold") resonator shows extreme field enhancements throughout various parts of the resonator (including its apex), the plasma-filled ("hot") resonator exhibits substantial field enhancements at the apex of the resonator, and at its left and right sides. Field distribution of optical fields has significant effect on the spatial localization of ablation. For example, for the example of $J_2 = 0.12\,J\,cm^{-2}$ fluence shown in Fig. 4c, we observe a large, keyhole-type ablated area near the microresonator apex and wing-like ejecta at its sides. Further reducing the on-axis per-pulse fluence to $J_3 = 0.09$ $J\,cm^{-2}$ reduces the ablation footprint as shown in Fig. 4d. Specifically, no wing-like structures are formed on the sides of the microresonator, and one of the narrowest trenches ($w \approx \lambda/80$, or $w \approx 50\,nm$) is formed. The high aspect ratio ($h/w \approx 12 : 1$) of the trench is achieved with a $N = 100$-pulse train.

Next, we comment on the rationale for using the MIR radiation for FLANEM. Choosing a laser with a longer wavelength may appear counter-intuitive if the objective is to produce the smallest features possible[11]. There are several reasons for using MIR light in the context of FLANEM. First, the size of the prefabricated resonant structures producing the initial optical energy localization scales with $\lambda_L$, i.e. they are relatively large and can be reliably produced using a variety of fabrication techniques. Second, the oscillatory energy $U_{osc}$ of an electron tunneled into the conduction zone scales as $\lambda_L^2$: $U_{osc} = q_e^2 E_L^2 / 4m\omega_L^2 \approx 9.3[eV] \times I_{hs}\left[10^{14}W\,cm^{-2}\right] \times \lambda_L^2[\mu m^2]$, where $I_{hs}$ is the MIR intensity at the hot spot. Therefore, assuming the incident laser intensity $I_L^{(in)} \sim 0.5 \times 10^{12}\,W\,cm^{-2}$ and the hot spot intensity enhancement $I_{hs} \sim 25 I_L^{(in)}$ corresponding to a 5-fold electric field enhancement, we obtain $U_{osc} \sim 18\,eV$, i.e. much larger that the electron

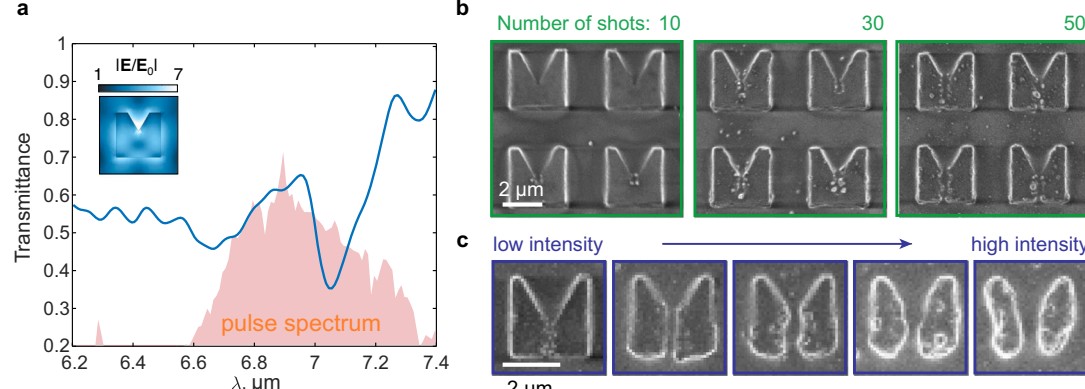

**Fig. 3 | Scalability of the trench formation process. a** Transmittance spectrum of a microresonator sample with upscaled dimensions (Si thickness is 600 nm). The resonance is observed around $\lambda_{res} \approx 7\,\mu m$, while the pump wavelength is $\lambda_L \approx 6.9\,\mu m$. The inset shows the local filed distribution within the half-height cross-section of the resonator. **b** Shot-controlled trench formation with the final width of -120 nm at $N = 50$ shots. **c** Intensity control of the trench width from 150 nm at $0.15\,J\,cm^{-2}$ to 550 nm at $1.2\,J\,cm^{-2}$.

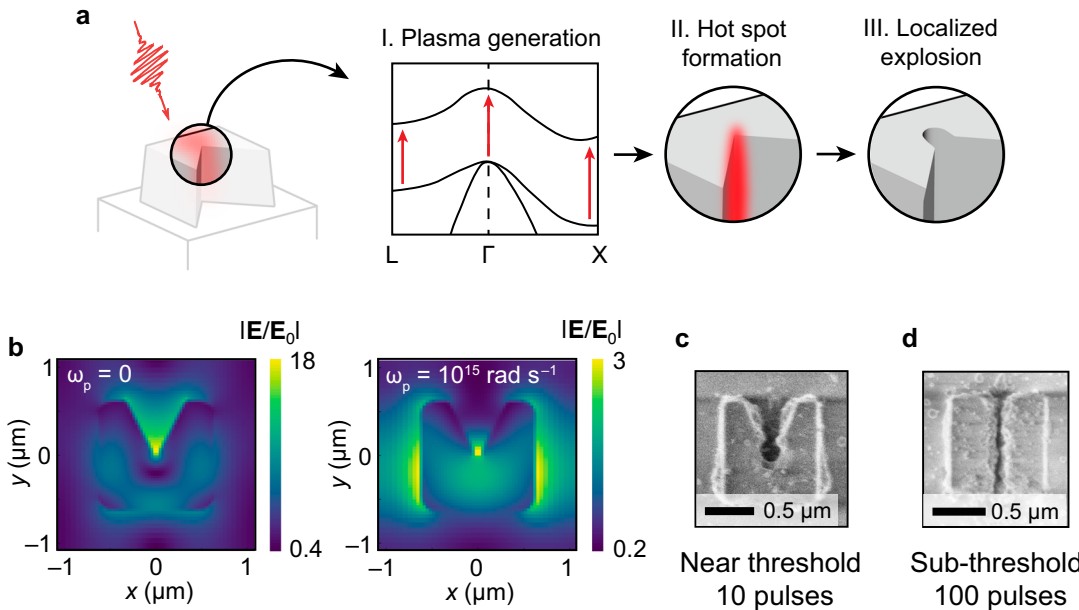

**Fig. 4 | Mechanism of plasma-assisted nanotrench formation. a** Incoming femtosecond-laser pulse couples to the eigenmode of the resonator, producing enhanced localized fields ("hot spots"). (I) The fields promote electron-hole plasma generation throughout the resonator. (II) The electron-hole plasma-filled resonator modifies the mode field to form a hot spot at its apex. (III) Coulomb and phase explosions eject material from the apex, seeding the trench and future ejection sites. **b** Electric field profiles of the unperturbed (left: $\omega_p = 0$) and plasma-filled

(right: $\omega_p = 10^{15}$ rad s$^{-1}$) resonators. In the latter case, the fields are pushed to modify the resonator at its apex, as well as the microresonator's right and left sidewalls. **c** The resulting 10-pulse modification of the resonator near the single-pulse damage threshold, showing a large keyhole-type orifice at the apex and partial ablation on the sides. **d** Example of a 100-pulse narrow (50 nm or $\lambda_L/78$) trench formation using a sub-threshold fluence of $J_3 = 0.09$ mJ cm$^{-2}$. Because of the small fluence used, reshaping only affects the area near the apex of the microresonator.

affinity ($W_{EA} \approx 4.05$ eV) or the bandgaps (indirect: $E_g^{(ind)} \approx 1.1$ eV, direct: $E_g^{(dir)} \approx 3.28$ eV) of silicon. Therefore, those electrons near the Si-air interface tunneling from the valence into the conduction zone can gain sufficient kinetic energy from the optical field to overcome the work-function barrier and exit the solid. Much higher laser intensities would be required for near-IR laser pulses to impart the same kinetic energy to the tunneled electrons. Third, because MIR photons with energies much smaller than the bandgap ($\hbar\omega_L \ll E_g$) satisfy the Keldysh adiabatic tunneling condition[41–43] $\gamma_K < 1$ (where $\gamma_K = \omega_L \sqrt{m_{eff} E_g}/q_e E_{hs}$ is the Keldysh adiabaticity parameter and $E_{hs} \propto \sqrt{I_{hs}}$ is the peak optical field at the hot spot), the volume of the high-density hot plasma produced by the laser pulse is much smaller than that of the optical energy itself – hence the cascading of the localization scales from hundreds of nanometers (the size of the photonic hot spot) to tens of nanometers (the size of the ablated region).

Finally, the post-tunneling electron displacement $\Delta_{hs} \sim q_e E_{hs}/m_{eff}\omega_L^2$ inside the hot spot also scales as $\lambda_L^2$: $\Delta_{hs}$[nm]$\approx (m_0/m_{eff})\lambda_L^2[\mu m^2]\sqrt{I_{hs}\left[10^{14}W\ cm^{-2}\right]}$. Assuming $m_0/m_{eff} \sim 3.3$ and $I_{hs} \sim 10^{13}$ W cm$^{-2}$ yields $\Delta_{hs} \sim 16$ nm, and with $m_0/m_{eff} \sim 5, \Delta_{hs} \sim 25$ nm. Therefore, all electrons promoted to the conduction zone and located within $\sim 20$ nm from the Si-air interface are expected to leave the solid. The above scaling offers further justification for using a longer-wavelength laser for FLANEM, while the numerical estimate is fully consistent with our experimentally measured ablation rate of $v_{abl} \approx 30$ nm per pulse.

A 3D simulation based on the EPOCH (Extendable PIC Open Collaboration) code[44] incorporated with the homebuilt refractive index and dynamic Keldysh photoionization modules[41–43] was performed to elucidate the nano-trenching process. The simulation resolution was optimized to capture the resonant modes of interest and the resulting

laser-material interactions. The parameters applied in the simulation are those of a polarized 200-fs laser pulse with a center wavelength of 3.9 µm and a peak fluence of 0.1 J cm$^{-2}$ is incident upon the resonator from the free space. This simulation only considers the direct inter-band excitation because the indirect excitation is negligible in the MIR regime[36].

As shown in Fig. 5a, the $|\mathbf{E}|^2$ distribution manifests localized enhancement at the notch apex and the two sides before the pulse peaks at the resonator surface ($t < 0$). Here, the brown contour corresponds to $E = 2.6$ V nm$^{-1}$ and $I = 3.6$ TW cm$^{-2}$, which is reasonably close to the experimental values given the resonator-provided local-field enhancement. After $t = 0$, near-solid-density plasmas trigger at the notch apex along the depth and keep growing during the rest of the pulse; see Supplementary Movies 1–6. This refractive index modification directly affects the resonant modes, so the $|\mathbf{E}|^2$ distribution becomes irregular but retains the intensification at the notch apex. The excited electrons are continually heated by this intensification and transfer the kinetic energy to ions via collisions, confirming the three-step model in Fig. 4a. Figure 5b shows that the ion temperature increases rapidly at the notch apex after $t = 0$. At 237 fs delay, Fig. 5b suggests that the rod-like melting front (1683 K) penetrates the resonator, consistent with the experimentally observed nano-trenching at a scale of around 100 nm. Importantly, this figure scales down with the downscaling of the parameters of the experiment; see Supplementary Fig. 1. Moreover, inside the ultrafast-melting "rod" is a boiling front (2628 K) in a similar shape but a smaller size, with a highest inner temperature up to $10^4$ K. This rapid and localized heating with the extreme temperature gradient will very likely lead to phase explosion[36,45,46] and explosive ejection of the superheated material[47]. Therefore, local electromagnetic field engineering leads to targeted material removal at a deep-subwavelength scale and various experimentally observed reshaping scenarios. Our PIC simulations also confirm the polarization sensitivity of FLANEM. Figure 5c, d reveal two scenarios of the electron density evolution for the E-field polarized at

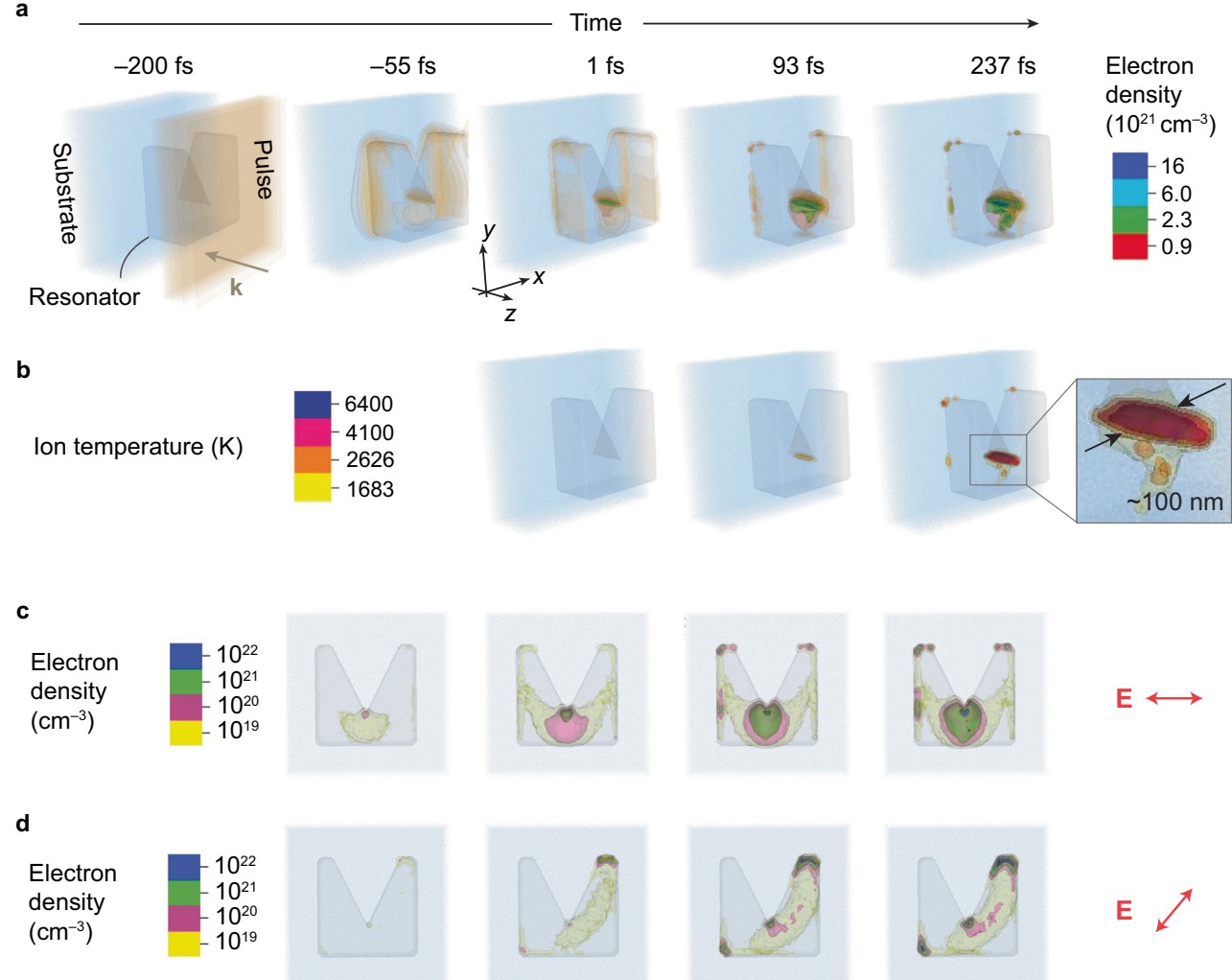

**Fig. 5 | Particle-in-cell simulation results. a** Temporal evolutions of the electric field (the brown contour corresponds to $E = 2.6$ V/nm except for $t = -200$ fs) and electron density (the red, green, turquoise, and blue contours) in the M-shaped resonators. The moment when the pulse peak reaches the resonator surface is defined as $t = 0$. **b** Temporal evolution of the lattice ion temperature, which starts to rapidly build up through electron collisions after the pulse has left the resonator. As a result, a rod-shaped phase-explosion region is formed, spanning the whole thickness of the *resonator*, and having a typical diameter of about 100 nm. **c** Temporal evolution of the electron density (log scale) at mid-height cross-section for a resonator excited by a horizontally polarized wave ($\theta = 0°$). A symmetric distribution of the electron generation initiates a downward trench. **d** Same for $\theta = 45°$. An asymmetric distribution of the electron density near the apex suggests trench formation perpendicular to the polarization of the incoming beam.

$\theta = 0°$ and $\theta = 45°$, respectively. In both cases, the apex shows a strong localized electron-hole pair generation, with the $\theta = 0°$ case giving a higher injection density. However, the general direction of the injection is different, oriented in both cases perpendicularly to the polarization plane, confirming that the polarization is an important degree of freedom in FLANEM.

Given the sub-picosecond time it takes to locally break the effective melting threshold, it is unlikely that thermal processes are responsible for the material removal from the apex of the resonator[48]. The phase explosion is a common outcome of femtosecond pulse ablation[49] which is characterized by ultrafast transfer of energy densities corresponding to equivalent lattice temperatures exceeding that of melting. The latter is confirmed by our PIC simulations, where the localized effective temperature far exceeds the melting temperature at timescales of less than 1 ps, therefore, before any pressure can be established by phonons. Another proposed mechanism, the Coulomb explosion, typically observed in Si at fluences about an order of magnitude above the damage threshold[16], requires the ionization of Si to Si$^+$ through photoeffect to over a critical density. While further studies, such as time-of-flight mass-spectrometry, will be required to confirm this route, we can conclude the following: (a) the estimated localized ponderomotive energy of the electrons is enough to escape the material and ionize it, and (b) the estimated hot-spot intensities are commensurate with those required for Coulomb explosions (with adjustments for the wavelength).

Our final remark is that further studies are needed to elucidate the role of the minimum feature size of the prefabricated microresonator on the nanostructuring capacity of FLANEM. Irradiating resonant particles of various shapes, apex sharpness, and electromagnetic mode distributions will establish the feasibility of our method for amending the existing photolithographic techniques where the spatial resolution is limited by the wavelength of light. Additionally, the material choice will define the explosion dynamics and spatial scales, which is an exciting avenue for follow-up studies.

In conclusion, we have demonstrated laser-induced nanoscale reshaping in silicon microresonators by mid-infrared femtosecond-laser pulses, resulting in feature sizes down to $\sim \lambda/80$ and aspect ratios of more than 10:1. Material removal manifests in the form of a

nanotrench, whose topography can be controlled by the number of pulses, their intensity, and polarization. The nanotrench propagates at approximately 30 nm per shot, spans the entire thickness of Si, and can be imposed on many resonators simultaneously. The trench formation is confirmed by particle-in-cell simulations and analytical estimations and is, as a plausible explanation, likely caused by highly localized Coulomb or phase explosions, followed by rapid ejection of superheated material. Given the designer nature of localized resonator fields, we envision that light-driven explosions will enable a new family of light-based nanofabrication approaches to see use in plasma physics, semiconductor processing, nanofluidics, and laser medicine.

## Methods

### Sample fabrication

The pattern was defined in a PMMA 495 A4 resist layer (baked for 12 min at 90 °C) by EBL with a 1000 μC cm$^{-2}$ dose and PEC correction (JEOL 9500FS), and developed in a MIBK: IPA solution (diluted 1:3), on a silicon-on-sapphire wafer (MTI Corp, purity 99.996%). A 30-nm layer of Cr was thermally evaporated, and the resist was then lifted off in acetone by a 12-min bath with sonication. The pattern was transferred to the device layer by HBr reactive ion etch (Oxford Instruments Cobra), and the remainder of the mask was removed in a wet Cr etch. Each sample was made large enough (from $600 \times 300\,\mu m^2$ to $690 \times 345\,\mu m^2$), with tens of thousands of identical microparticles, for each irradiation event to utilize a fresh spot.

### Sample characterization

The fabricated samples before and after laser irradiation were characterized using a Zeiss Supra Scanning Electron Microscope operated at 1 kV without a discharge layer. The infrared spectra were obtained using a Bruker Vertex 70 spectrometer with an external setup that used a weakly converging beam focused to about a $200 \times 200\,\mu m^2$ spot with a ZnSe lens.

### Laser damage experiments

$\lambda_L = 3.9\,\mu m$ experiments: The M-shaped resonators were irradiated with pulses from a homebuilt KNbO$_3$/KTA 3-crystal/3-pass optical parametric amplifier pumped by a 80-fs Ti: Sapphire chirped pulse amplification system with a central wavelength of 780 nm and pulse energy of 4 mJ. The repetition rate of the laser system was set by the user-controlled Pockels cell to be 10 Hz, which allowed pulse picking with a mechanical shutter (Melles Griot IES). The number of pulses sent to the metasurface varied from 1 to 100. For each irradiation, a fresh spot on the sample was chosen. The spatial step for each new spot was at least $50\mu$ m, which is larger than the typical beam fwhm of $36\mu$ m. The beam was profiled in its focal spot using a MIR profiler (DataRay WinCamD). The beam's polarization was controlled with a broadband MIR half-waveplate, and the pulse power was controlled with a pair of wire-grid polarizers and monitored with a calibrated biased PbSe photodetector.

$\lambda_L = 7\,\mu m$ experiments: The experiments were performed at the Advanced Laser Light Source (ALLS, Varennes, QC, Canada). A Ti: Sapphire pump laser was used to drive a high-energy OPA line ending with a difference frequency generation stage to create a beam at 7 μm, 50 Hz, a 500 nm FWHM spectral bandwidth, and up to 50 μJ per pulse. Spectral characterization was done on a Spectral Products gratings-based slit monochromator with a liquid nitrogen-cooled HgCdTe detector. DC servos and shutters from Thorlabs were used to control the number of pulses hitting the sample. For focusing, a ZnSe lens with 50 mm focal length was used. Polarization and power were controlled with a wire-grid polarizer and a half-waveplate. The beam waist was measured to be 40 μm by leaving the shutter open at high power and completely ablating the substrate, resulting in an easily identifiable area to measure optically.

### Particle-in-cell simulations

We used EPOCH code to model the interaction between the MIR pulse and the metasurface. The simulation space is a $2.13\,\mu m(x) \times 2$ .$13\,\mu m(y) \times 3\,\mu m(z)$ cuboid, in which the $z$ axis is parallel with the propagation direction of the laser pulse. The laser pulse comes along the $z$ axis from the right boundary and is absorbed in the left. The intensity temporal profile is set to be cosine squared. The 0.6 μm-thick silicon resonator is centered at the $x$-$y$ plane with the four outer sides of 1.21 μm. The outer four corners are chamfered to be consistent with the experiments. The notch of the resonator is a symmetrical triangle with both altitude and base of 0.83 μm. To model the resonance modes caused by the metasurface consisting of many micro-resonators, the $x$–$z$ and $y$–$z$ boundaries are set to be periodic. These imposed periodic conditions imply that the laser spot size is much larger than the simulation space; hence a constant spatial amplitude was applied. The substrate is a 0.9-μm-thick Al$_2$O$_3$ slab. The ionization is suppressed for the substrate. The spatial resolution is $21.3nm(x) \times 21.3nm(y) \times 25nm(z)$. Initially, the temperatures of the ambient environment and the target is 300 K. In the resonator, each unit cell has 2048 macroparticles with each representing 275 Si atoms (the molecule density of Si is $5.01 \times 10^{22}$ cm$^{-3}$). The wavelength-independent refractive index of Si is 3.43, and the refractive index of Al$_2$O$_3$ is 1.68, the average of ordinary and extraordinary refractive indices. The homebuilt Keldysh photoionization and the built-in collision modules are applied to simulate the electronic dynamics in the silicon resonators. The optical effective mass is 0.3 $m_e$[50], in which $m_e$ is the free electron mass; the (direct) bandgap is 3.4 eV[36], the Coulomb logarithm is 15. The total simulation time was 440 fs with $t = 273$ fs as the terminal time.

Modeling the photoionization process in Si is challenging due to its complex band structure associated with multiple direct and indirect inter-band transitions. However, the previous work suggests that the indirect transition becomes increasingly unimportant from the near-IR to MIR regime, so it is reasonable to neglect its contribution at the wavelength of 3.9 μm. Moreover, Ref. [50] points out that, among the three direct transition channels, $\Gamma$ point (3.28–3.5 eV), $L$ valley (3.3 eV), and $X$ valley (4.27 eV), the contribution to total photoionization via $L$ valley is several orders lower than those via $\Gamma$ point and $X$ valley. Compared to the transition through $\Gamma$ point, the bandgap between the $X$ valley and conduction band is much higher, and the effective hole and electron masses are also higher. Accordingly, the photoionization rate will be much lower. Therefore, our treatment for the electron excitation through $\Gamma$ point is reasonable and should capture most of the physics of nanotrench formation.

## Data availability

The data that support the findings of this study are available within the paper and the Supplementary Information. Other relevant data are available from the corresponding authors on request.

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

## Acknowledgements

The authors acknowledge stimulating discussions with N. Bulgakova. Support from the Office of Naval Research (Award No. N00014-22-1-2199) and the Air Force Office for Scientific Research (Award Nos. FA9550-21-1-0421 and FA9550-20-1-0278) is acknowledged. This work was performed in part at the Cornell NanoScale Facility, a member of the National Nanotechnology Coordinated Infrastructure (NNCI), which is supported by the National Science Foundation (Grant NNCI-2025233). The PIC simulations were performed at the Ohio Supercomputer Center (OSC). The ALLS facility was funded and supported by the Canada

Foundation for Innovation (CFI), INRS, and Ministère de l'Économie, de l'Innovation et de l'Énergie (MEIE) from Québec. This work was supported by the U.S. Department of Energy Office of Science, Fusion Energy Sciences under Contract No. DE-SC0021246: the LaserNetUS initiative at the Advanced Laser Light Source. All the authors acknowledge LaserNetUS for the beamtime provided under proposal K209.

## Author contributions

M.R.S. and G.Sartorello designed the experiments. M.R.S., G.Sartorello, and M.B. fabricated the samples and performed FTIR and SEM observations. S.Z., J.S., and E.C. performed PIC calculations. M.R.S., G.Sartorello, M.T., N.T., A.A., J.B., M.B., S.L., F.L., and E.C. performed laser modification experiments. E.C. and G.Shvets supervised the project. All the authors analyzed the data and contributed to the final manuscript.

## Competing interests

The authors declare no competing interests.
