## [Peer Review File · Nature Communications]

Nanoscale reshaping of resonant dielectric microstructures by light-driven explosionsREVIEWER COMMENTS

Reviewer #1 (Remarks to the Author):

Shcherbakov and co-authors study trench formation in M-shaped microresonators illuminated with femtosecond laser pulses in the MIR range. They use $3.9\ \mu\text{m}$ wavelength and illuminate pre-fabricated nanostructures, which makes the paper less interesting for practical applications but more interesting from the physical point of view. The paper describes very interesting and new physical effects, but it is not written very clearly and some interesting aspects were omitted by the authors. However, it can be improved if the authors address the following points:

1. It was not clear for me what are the resonators made of and where exactly you make the trenches: If the resonators are made of Si, why do you say that they are insulating (Si is semiconductor with relatively high conductivity at the room temperature)? Or the resonators are covered with SiO₂? Or they are made of PMMA, but in this case where is the 600 nm Si layer?
2. In the introduction you mention different mechanisms of the surface modification (lines 55-56). Can you clarify the difference between (a) Coulomb explosion, (b) phase explosion, (c) lattice destabilization, (d) non-thermal melting.
3. The authors call the method FLANEM. I think that 20 years ago it was called near-field ablation. Can you please either discuss the difference or use the traditional name.
4. The statement that the threshold fluence of $0.1\ \text{J}/\text{cm}^2$ is lower than typically used for flat Si (line 111-112) is not correct. In the visible and NIR range the threshold fluence is even lower. Or you mean the MIR range?
5. In the third step of the theory the trench formation is attributed to the "coulomb or phase explosion". I see no evidence for exactly this mechanism in the paper. If this is an assumption, please soften this statement.
6. The discussion about the smallest possible features through the paper awakes the impression that this method can be used as a universal method for laser material processing. But these trenches can be made only in the pre-fabricated M-shaped resonators at the given dimensions and the wavelength. Or you can demonstrate formation of these trenches in a broad range of microstructures?
7. How the shape of the resonator (length, height, angles in the corners) affect the electron density and temperature at the hot spot? Can you give a prediction for different shapes of the microstructures or at least estimate how they should change if commonly-used NIR- or visible fs lasers are used?
8. If the laser processing is done at ambient conditions, the surface of the silicon structures must be coated with an oxide layer. How the oxide affects the numerical simulations and the analytical estimations? Or you remove the oxide and keep the sample without oxygen all the time?

Reviewer #2 (Remarks to the Author):

In this work, the author reports femtosecond-laser-assisted deeply-subwavelength features in silicon, by localized laser-induced phase explosions in pre-fabricated silicon resonators. Using short trains of mid-infrared laser pulses, the author demonstrates the controllable formation of high aspect ratio nano trenches. Furthermore, the trench geometry can be well controlled by multiple parameters of the laser pulse train. Particle-in-cell simulations reveal localized heating of silicon beyond its boiling point and suggest its subsequent phase

explosion on the nanoscale commensurate with the experimental data. In general, the results are novelty and match the theoretical calculation. In this regard, I would suggest that this paper is accepted for publication after minor revisions. Some comments are reported below:

1. The author proposed a femtosecond pulsed laser-assisted micro/nano machining method and the research results are all based on silicon. As a new technology, can this method be implemented in other material systems, such as GaN, SiC, or other 2D materials?
2. The laser etching process will produce a lot of heat and lead to a high temperature, will this affect the properties of the processed material? The authors should supplement relevant experimental data.

REVIEWER COMMENTS

Reviewer #1 (Remarks to the Author):

Shcherbakov and co-authors study trench formation in M-shaped microresonators illuminated with femtosecond laser pulses in the MIR range. They use $3.9\ \mu\text{m}$ wavelength and illuminate pre-fabricated nanostructures, which makes the paper less interesting for practical applications but more interesting from the physical point of view.

The paper describes very interesting and new physical effects, but it is not written very clearly and some interesting aspects were omitted by the authors.

Our response:

We thank the Reviewer for their time dedicated to reading and for suggestions to improve our manuscript. We are pleased that the reviewer finds “interesting and new physical effects” in our observation.

Reviewer #1:

However, it can be improved if the authors address the following points:

1. It was not clear for me what are the resonators are made of and where exactly you make the trenches: If the resonators are made of Si, why do you say that they are insulating (Si is semiconductor with relatively high conductivity at the room temperature)? Or the resonators are covered with SiO₂? Or they are made of PMMA, but in this case where is the 600 nm Si layer?

Our response:

Thank you for pointing out the lack of clarity in our description of the resonator material. The resonators are made of intrinsic silicon etched in the device layer of a silicon-on-sapphire wafer. We have revised Figure 1 to indicate the material stack unambiguously.

As for the insulating nature of Si, it is not claimed anywhere in the manuscript. We call our microresonators “dielectric,” because the residual conductivity of intrinsic silicon is negligible in the spectral range of interest (mid-infrared). The response and near-field mode distributions are equivalent to those of a dielectric material with a purely real refractive index of $n \approx 3.4$. While we acknowledge the residual low-frequency conductivity of intrinsic silicon, it is widely accepted by the community that it can be considered dielectric in the wide range of frequencies above the plasma frequency [A1-3].

[A1] Jahani, Jacob, *Nature Nanotechnology* **11**, 23–36 (2016).

[A2] Liu et al., *Advanced Optical Materials* **4**, 1457-1462 (2016).

[A3] Kuznetsov et al., *Science* **354**, aag2472 (2016).

Changes in the manuscript:

Material designations have been duplicated in the inset of Fig. 1, and the material designations were made clearer.

Reviewer #1:

2. In the introduction you mention different mechanisms of the surface modification (lines 55-56). Can you clarify the difference between (a) Coulomb explosion, (b) phase explosion, (c) lattice destabilization, (d) non-thermal melting.

Our response:

We believe the confusion here stemmed from the phrase “lattice destabilization, and non-thermal melting” whereby the former is often a part of the latter. In the way this is phrased, the comma emphasized the disparate nature of these processes, which is not entirely correct. Moreover, phase explosion is typically categorized as non-thermal (over)heating process. We have therefore rephrased the sentence which, in its revised form, is unambiguous.

Changes in the manuscript:

Text changed (p. 3): “Several non-thermal mechanisms of surface modification have been identified,^{13–18} ...”

Reviewer #1:

3. The authors call the method FLANEM. I think that 20 years ago it was called near-field ablation. Can you please either discuss the difference or use the traditional name.

Our response:

We thank the reviewer for bringing up the important point of a similar effect reported in prior literature, included in those we cite in our original reference list. We are convinced that these effects are different in their nature.

The effects the reviewer refers to have traditionally been associated with the near-field field enhancement brought by plasmonic nanoresonators; see refs. [A4,A5]. To recap, subwavelength plasmonic nanoparticles can concentrate the impinging electromagnetic field in nanoscale areas of space.^{A1,A2} However, the linear nature of hot-spot formation in metallic particles reduces the expected localization of ablated material, reaching inscribed features to $\sim \lambda_L/10$ in lateral dimensions, whereas the highly nonlinear nature of FLANEM enables almost an order of magnitude improvement in this figure.

[A4] Plech et al., Femtosecond laser near field ablation, *Laser Photon Rev* **3**, 435-451 (2009).

[A5] Plech et al., Femtosecond laser near-field ablation from gold nanoparticle, *Nat Phys* **2**, 44–47 (2006).

Changes in the manuscript:

Text added (pp. 5-6): "Similarly, FLANEM is distinct from near-field ablation enabled by subwavelength plasmonic nanoparticles.^{34,35} The linear nature of hot-spot formation in metallic particles reduces the expected localization of ablated material, reaching inscribed features to $\sim\lambda_L/10$ in lateral dimensions, whereas the highly nonlinear nature of FLANEM enables almost an order of magnitude improvement in this figure."

Reviewer #1:

4. The statement that the threshold fluence of 0.1 J/cm² is lower than typically used for flat Si (line 111-112) is not correct. In the visible and NIR range the threshold fluence is even lower. Or you mean the MIR range?

Our response:

While the Reviewer is correct in stating the damage threshold of Si is lower in the visible and near-IR, the experimentally confirmed phase explosion thresholds at the wavelength of choice (3.9 μm) is about 0.2-0.3 J/cm², as experimentally reported in our previous publication.^{A6}

[A6] Werner, K. et al. Single-Shot Multi-Stage Damage and Ablation of Silicon by Femtosecond Mid-infrared Laser Pulses. *Scientific Reports* **9**, 19993 (2019).

Reviewer #1:

5. In the third step of the theory the trench formation is attributed to the "coulomb or phase explosion". I see no evidence for exactly this mechanism in the paper. If this is an assumption, please soften this statement.

Our response:

The following observations lead to the stated explanation of the reshaping being either Coulomb or phase explosion.

The phase explosion is a common outcome of femtosecond pulse ablation [A7] which is characterized by ultrafast transfer of energy densities corresponding to equivalent lattice temperatures exceeding that of melting. The latter is confirmed by our PIC simulations, where the localized effective temperature far exceeds the melting temperature at timescales of less than 1 ps, therefore, before any pressure can be established by phonons.

Coulomb explosion, typically observed in Si at fluences about an order of magnitude above the damage threshold [A8], requires the ionization of Si to Si⁺ through photoeffect to over the critical density. While further studies, such as time-of-flight mass-spectrometry, will be required to confirm this route, we can conclude the following: (a) the estimated localized ponderomotive energy of the electrons is enough to escape the material and ionize it, and (b)

the estimated hot-spot intensities are commensurate with those required for Coulomb explosions (with adjustments for the wavelength).

Given the time it took to break the localized effective melting “temperature,” it is highly unlikely that thermal processes are responsible for the material removal from the apex of the resonator. However, we agree with the reviewer that the statement can be softened. We have updated the text accordingly.

[A7] Shugaev, M. V. et al. Laser-Induced Thermal Processes: Heat Transfer, Generation of Stresses, Melting and Solidification, Vaporization, and Phase Explosion. in Handbook of Laser Micro- and Nano-Engineering (ed. Sugioka, K.) 83–163 (Springer International Publishing, 2021)

[A8] Roeterdink, W. G. et al. Coulomb explosion in femtosecond laser ablation of Si(111). *Appl. Phys. Lett.* **82**, 4190–4192 (2003)

Changes in the manuscript:

Text added (p. 16): “Given the sub-picosecond time it takes to locally break the effective melting threshold, it is unlikely that thermal processes are responsible for the material removal from the apex of the resonator.⁴⁸ The phase explosion is a common outcome of femtosecond pulse ablation⁴⁹ which is characterized by ultrafast transfer of energy densities corresponding to equivalent lattice temperatures exceeding that of melting. The latter is confirmed by our PIC simulations, where the localized effective temperature far exceeds the melting temperature at timescales of less than 1 ps, therefore, before any pressure can be established by phonons. Another proposed mechanism, the Coulomb explosion, typically observed in Si at fluences about an order of magnitude above the damage threshold,¹⁶ requires the ionization of Si to Si⁺ through photoeffect to over a critical density. While further studies, such as time-of-flight mass-spectrometry, will be required to confirm this route, we can conclude the following: (a) the estimated localized ponderomotive energy of the electrons is enough to escape the material and ionize it, and (b) the estimated hot-spot intensities are commensurate with those required for Coulomb explosions (with adjustments for the wavelength).”

Text changed (p. 17): “...is, as a plausible explanation, likely caused by highly localized Coulomb or phase explosions.”

Text changed (p. 5): “... we conjecture that the resulting nanoscale ablation of Si is attributed to highly localized Coulomb or phase explosions”

Reviewer #1:

6. The discussion about the smallest possible features through the paper awakes the impression that this method can be used as a universal method for laser material processing. But these trenches can be made only in the pre-fabricated M-shaped resonators at the given dimensions and the wavelength. Or you can demonstrate formation of these trenches in a broad range of microstructures?

Our response:

The proposal of using our method as a universal nanofabrication method at this time, although exciting, is contingent upon the available resonant profiles and the subwavelength localization of electromagnetic field hot spots. Any microstructure possessing subwavelength localization, combined with the highly nonlinear nature of the material removal process, can support deep-subwavelength trench formation. Examples of such structures can be nanoparticles of spherical, elliptical, or disk shapes, wires, and other simple resonant geometries. While beyond the scope of the current report, we believe that further investigations will elucidate the role of the minimum particle feature size in FLANEM.

Changes in the manuscript:

Text added (p 16): “Our final remark is that further studies are needed to elucidate the role of the minimum feature size of the prefabricated microresonator on the nanostructuring capacity of FLANEM. Irradiating resonant particles of various shapes, apex sharpness, and electromagnetic mode distributions will establish the feasibility of our method for amending the existing photolithographic techniques where the spatial resolution is limited by the wavelength of light.”

Reviewer #1:

7. How the shape of the resonator (length, height, angles in the corners) affect the electron density and temperature at the hot spot? Can you give a prediction for different shapes of the microstructures or at least estimate how they should change if commonly-used NIR- or visible fs lasers are used?

Our response:

We agree with the reviewer in that the importance of the resonator shape and dimensions cannot be overstated. Ultimately, the dimensions of the resonator determine the central wavelength of the electromagnetic mode that drives FLANEM. To illustrate this further, we have run **additional experiments and simulations** showing settings that hosts a scaled-down or scaled-up version of our initial experiment.

The new Figure 3 of the manuscript shows a set of data where we have up-scaled the dimensions of the experiment. The newly fabricated samples have proportionally larger dimensions that show a resonance at $\lambda_{res} = 7 \mu\text{m}$. We have successfully repeated our experiment at this wavelength using our collaborators' laser facility and proved that at this wavelength, trenches down to $\lambda/60$ in width are possible.

The new Supplementary Figure 1 shows a set of PIC calculations for a different scenario, where the dimensions of the structure have been halved, enabling a resonance at $\lambda_{res} = 1.95 \mu\text{m}$. In this simulation, the hot spot of the overheated ion lattice has a characteristic spatial scale of 40 nm, which is around half of what the original simulations predicted, showing promising scalability toward the smaller feature sizes in FLANEM. Shorter wavelengths, e.g., 800 nm or 1,040 nm, cannot be reasonably simulated in our structure because of the onset of indirect-gap

single-photon absorption that is not included in our model. Nevertheless, materials with larger band gaps may be a promising avenue for the further reduction of FLANEM's feature sizes.

Changes in the manuscript:

Added new Figure 3 with a caption and main text description, showing up-scaled experiments.
Added new Figure S1 in the supplementary material showing down-scaled PIC results.

Reviewer #1:

8. If the laser processing is done at ambient conditions, the surface of the silicon structures must be coated with an oxide layer. How the oxide affects the numerical simulations and the analytical estimations? Or you remove the oxide and keep the sample without oxygen all the time?

Our response:

The experiments were conducted at ambient conditions. Therefore, the existence of the native oxide layer was assumed. However, the thickness of the native oxide (~2 nm) has historically been negligible compared to the bulk of the microresonators with their characteristic dimensions in hundreds of nanometers. Since the wavelength of the resonance is, as an approximation, proportional to the $n_{\text{eff}} - 1$, where n_{eff} is the effective refractive index of the resonator. The overall change of the resonator central wavelength due to the native oxide can be roughly estimated as $\frac{\Delta\lambda}{\lambda} \approx \frac{\Delta n_{\text{eff}}}{n_{\text{eff}} - 1}$, where $\Delta n_{\text{eff}} \approx n_{\text{Si}} - n_{\text{Si+SiO}_2}$, $n_{\text{Si}} = 3.4$, and $n_{\text{Si+SiO}_2}$ is the adjusted effective refractive index due to the addition of the native oxide, estimated at $n_{\text{Si+SiO}_2} = (V_{\text{Si}}n_{\text{Si}} + V_{\text{SiO}_2}n_{\text{SiO}_2}) / (V_{\text{Si}} + V_{\text{SiO}_2})$, where V is the respective volume of the material. At 2 nm thickness of SiO₂ and approximating the microresonator as a cuboid with 600 nm x 1 um x 1 um dimensions, the ratio of the volumes $\frac{V_{\text{SiO}_2}}{V_{\text{Si}}} \approx 0.015$, and $n_{\text{SiO}_2} = 1.5$, the overall n_{eff} correction due to the oxide is $\Delta n_{\text{eff}} \approx 0.03$, making the central wavelength correction $\Delta\lambda \approx 50$ nm, which is well within the bandwidths of the pulse and the resonance. Additionally, note that the spectroscopic measurements prior to laser irradiation were conducted at ambient conditions, too, making the experimental resonance position already account for the possible wavelength adjustment.

Changes in the manuscript:

Text added (SI p.3):

Supplementary Section 1. The Role of Native Oxide

Since the experiments were conducted at ambient conditions, the existence of the native oxide layer on silicon was presumed. However, we render the role of the native oxide (thickness ~2 nm) negligible compared to the bulk of the microresonators with their characteristic dimensions. Since the wavelength of the resonance is, as an approximation, proportional to the $n_{\text{eff}} - 1$, where n_{eff} is the effective refractive index of the resonator. The overall change of the

resonator mode's central wavelength due to the native oxide can be roughly estimated as $\frac{\Delta\lambda}{\lambda} \approx \frac{\Delta n_{\text{eff}}}{n_{\text{eff}}-1}$, where $\Delta n_{\text{eff}} \approx n_{\text{Si}} - n_{\text{Si+SiO}_2}$, $n_{\text{Si}} = 3.4$, and $n_{\text{Si+SiO}_2}$ is the adjusted effective refractive index due to the addition of the native oxide, estimated at $n_{\text{Si+SiO}_2} = (V_{\text{Si}}n_{\text{Si}} + V_{\text{SiO}_2}n_{\text{SiO}_2})/(V_{\text{Si}} + V_{\text{SiO}_2})$, where V is the respective volume of the material. At 2 nm thickness of SiO₂ and approximating the microresonator as a cuboid with 0.6 μm x 1 μm x 1 μm dimensions, the ratio of the volumes $\frac{V_{\text{SiO}_2}}{V_{\text{Si}}} \approx 0.015$, and $n_{\text{SiO}_2} = 1.5$, the overall n_{eff} correction due to the oxide is $\Delta n_{\text{eff}} \approx 0.03$, making the central wavelength correction $\Delta\lambda \approx 50$ nm, which is well within the bandwidths of the pulse and the resonance.

Reviewer #2 (Remarks to the Author):

In this work, the author reports femtosecond-laser-assisted deeply-subwavelength features in silicon, by localized laser-induced phase explosions in pre-fabricated silicon resonators. Using short trains of mid-infrared laser pulses, the author demonstrates the controllable formation of high aspect ratio nano trenches. Furthermore, the trench geometry can be well controlled by multiple parameters of the laser pulse train. Particle-in-cell simulations reveal localized heating of silicon beyond its boiling point and suggest its subsequent phase explosion on the nanoscale commensurate with the experimental data. In general, the results are novelty and match the theoretical calculation. In this regard, I would suggest that this paper is accepted for publication after minor revisions.

Our response:

We thank the reviewer for their time evaluating our manuscript. We are pleased that the reviewer finds our results novel and acceptable for publication in Nature Communications after minor revisions.

Reviewer #2:

Some comments are reported below:

1. The author proposed a femtosecond pulsed laser-assisted micro/nano machining method and the research results are all based on silicon. As a new technology, can this method be implemented in other material systems, such as GaN, SiC, or other 2D materials?

Our response:

The point that the reviewer raised is very important. Our method can be universally applied to any semiconductor and dielectric material where the Keldysh impact ionization model is valid, establishing the high predictive power of our PIC simulations. The FLANEM mechanism relies on the multiphoton or tunneling free carrier generation mechanism observed to result in the

ablation of a variety of materials, including GaN, SiC, and 2D materials such as MoS₂ [A9] and others [A10]. We have added the relevant discussion to the main text.

[A9] Solomon et al., *Scientific Reports* **12**, 6910 (2022).

[A10] Kollipara et al., *Research* 2020, 6581250 (2020).

Changes in the manuscript:

Added in the main text (p. 16): “Additionally, the material choice will define the explosion dynamics and spatial scales, which is an exciting avenue for follow-up studies.”

Reviewer #2:

2. The laser etching process will produce a lot of heat and lead to a high temperature, will this affect the properties of the processed material? The authors should supplement relevant experimental data.

Our response:

In our particle-in-cell calculations, we have estimated the amount of heat generated through the relevant processes of free carrier generation and thermalization with the lattice. Fig. 5 shows the overall lattice temperature elevation and its distribution throughout the volume of the material, peaking at around 6,000–7,000 K in the hot spot. In this dynamic picture, the elevated temperature profile will result in the material removal from the V groove in the microresonator. However, when averaged over the material’s volume, the resonator’s overall temperature rise is fairly mild. With the hot spot volume being roughly $0.1 \mu\text{m} \times 0.1 \mu\text{m} \times 0.6 \mu\text{m} = 0.006 \mu\text{m}^3$, and the resonator volume estimated at $1 \mu\text{m} \times 1 \mu\text{m} \times 0.6 \mu\text{m} = 0.6 \mu\text{m}^3$, the averaged overall temperature rise of the resonator is estimated around 60-70 K which is far below the melting point of silicon (1,683 K), without taking into account the heat dissipation in the substrate and the removal of heat by the ejected particulates. Therefore, our fine-tuning of the laser beam intensity allowed us to highly localize material modification and removal without any significant modification to the remainder of the resonator, as confirmed by our experimental data. The studies of the microscopic changes, such as localized amorphization of Si, represent an exciting avenue for research that will be considered for our future studies.

We have updated the **Supplementary Information** text to reflect this discussion; see Supplementary Section 2.

REVIEWERS' COMMENTS

Reviewer #1 (Remarks to the Author):

Thank you for your detailed answering my comments and questions. Now I can recommend the paper for publication.

Reviewer #2 (Remarks to the Author):

The author responded well to the review comments and there are no further questions here.